# Genus *Ribes*: *Ribes aureum*, *Ribes pauciflorum*, *Ribes triste*, and *Ribes dikuscha*—Comparative Mass Spectrometric Study of Polyphenolic Composition and Other Bioactive Constituents

**DOI:** 10.3390/ijms251810085

**Published:** 2024-09-19

**Authors:** Mayya P. Razgonova, Muhammad Amjad Nawaz, Andrey S. Sabitov, Kirill S. Golokhvast

**Affiliations:** 1N.I. Vavilov All-Russian Institute of Plant Genetic Resources, B. Morskaya 42-44, Saint-Petersburg 190000, Russia; andrsabitov@rambler.ru (A.S.S.); golokhvast@sfsca.ru (K.S.G.); 2Advanced Engineering School, Far Eastern Federal University, Sukhanova 8, Vladivostok 690950, Russia; 3Advanced Engineering School (Agrobiotek), National Research Tomsk State University, Lenin Ave, 36, Tomsk 634050, Russia; 4Center for Research in the Field of Materials and Technologies, Tomsk State University, Lenin Ave, 36, Tomsk 634050, Russia; 5Siberian Federal Scientific Centre of Agrobiotechnology RAS, Centralnaya 2b, Presidium, Krasnoobsk 633501, Russia

**Keywords:** *Ribes*, *Ribes aureum*, *Ribes pauciflorum*, *Ribes triste*, *Ribes dikuscha*, tandem mass spectrometry, polyphenols, metabolome

## Abstract

This study presents the metabolomic profiles of the four *Ribes* species (*Ribes pauciflorum* Turcz., *Ribes triste* Pall., *Ribes dicuscha* Fisch., and *Ribes aureum* Purch.). The plant material was collected during two expeditions in the Russian Far East. Tandem mass spectrometry was used to detect target analytes. A total of 205 bioactive compounds (155 compounds from polyphenol group and 50 compounds from other chemical groups) were tentatively identified from the berries and extracts of the four *Ribes* species. For the first time, 29 chemical constituents from the polyphenol group were tentatively identified in the genus *Ribes*. The newly identified polyphenols include flavones, flavonols, flavan-3-ols, lignans, coumarins, stilbenes, and others. The other newly detected compounds in *Ribes* species are the naphthoquinone group (1,8-dihydroxy-anthraquinone, 1,3,6,8-tetrahydroxy-9(10H)-anthracenone, 8,8′-dihydroxy-2,2′-binaphthalene-1,1′,4,4′-tetrone, etc.), polyhydroxycarboxylic acids, omega-3 fatty acids (stearidonic acid, linolenic acid), and others. Our results imply that *Ribes* species are rich in polyphenols, especially flavanols, anthocyanins, flavones, and flavan-3-ols. These results indicate the utility of *Ribes* species for the health and pharmaceutical industry.

## 1. Introduction

A growing body of nutritional and pharmacological evidence links a diet rich in fruits and vegetables to a reduced risk of cardiovascular disease, cancer, diabetes, and other severe chronic diseases [1,2]. The main benefit of such a diet may be the increased intake of antioxidants, including carotenoids, tocopherols, and phenolic compounds [3]. These compounds are present in a wide range in the berries and leaves of the genus *Ribes*. Blackcurrant berries are an excellent source of these biologically active components (anthocyanins, flavone group, catechins, procyanidins, and phenolic acids). Previous studies have shown that blackcurrant is a good source of bioactive polyphenols (500–1342 mg/100 g total polyphenols), mainly anthocyanins [4]. In general, among the entire phenolic fraction, flavonoids are potent antioxidants in vitro and include compounds such as flavones, isoflavones, flavanones, catechins, and red, blue, and violet pigments known as anthocyanins [4,5,6]. Compounds other than vitamin C have also been shown to be major contributors to the antioxidant capacity of fruits [7]. Various studies have reported that blackcurrant (*Ribes nigrum* L.) is a rich source of dietary anthocyanins and antioxidants [8,9]. Numerous studies in recent years have shown that anthocyanins exhibit a wide range of biological activity, including both neuroprotective effects and antioxidant, antimicrobial, and anticarcinogenic activity [10,11].

Recently, the attention of the scientific community has been focused on the anti-inflammatory activity of anthocyanins [12]. Anthocyanins are known to act as antioxidants, but, in addition to their main property, they can interrupt or reverse the process of carcinogenesis by affecting intracellular signaling molecules involved in initiating and/or promoting cancer development. The effects of anthocyanin complex exposure appear to be cell type- and dose-dependent. Depending on their specific structure, anthocyanins influence various cellular signaling elements that are critical for the regulation of cell proliferation [13]. For example, the study by Tsuda et al. demonstrated a significant change in the expression of adipocytokines in human adipocytes treated with anthocyanins [14]. Based on the gene expression profile, increased levels of adiponectin and decreased levels of plasminogen activator inhibitor-1 and interleukin-6 were shown. This study showed that anthocyanins can regulate adipocytokine gene expression, improving adipocyte function associated with obesity and diabetes. Some anthocyanins are able to lower blood glucose levels and reverse the decline in pancreatic beta cells more effectively than glimepiride, a known insulin-secretory agent [15]. Thus, the dietary consumption of polyphenol-rich foods may be beneficial in preventing the onset of type 2 diabetes.

About 160 species constitute the genus *Ribes* within the Grossulariaceae family. Of the many species of *Ribes* known to exist in Russia, four species are found mostly in the Magadan region: *Ribes dikuscha*, *Ribes pauciflorum*, *Ribes triste*, and *Ribes aureum* [16]. The majority of uses for *Ribes* species are in ethnomedicine in China and Russia. Their efficacy in treating a multitude of ailments, such as hepatitis, arthritis, and joint pain, has been shown in numerous research works. This is a result of their beneficial health effects, which include antioxidant and anti-inflammatory qualities [17]. Most research work performed in the metabolomic research on *Ribes* species is dedicated to *Ribes nigrum* [18,19,20], whereas other species are less explored. Moreover, given their significance in the food, pharmaceutical, and health industries, it is necessary to further the exploratory biochemical research and find novel compounds. In this regard, four *Ribes* species were collected during expeditions to the Magadan region (Russian Federation) in 2023 and 2024. This study explores the metabolomic composition—specifically, the composition of the polyphenolic group—of the four *Ribes* species by mass spectrometry.

## 2. Results

### 2.1. Global Metabolome Profile of Berries of Four Ribes Species

The structural identification of each compound was performed on the basis of their accurate mass and MS/MS fragmentation by HPLC–ESI–ion trap–MS/MS. A total of 205 chemical compounds were identified from extracts of the four *Ribes* species (*R. pauciflorum*, *R. triste*, *R. dicuscha*, *R. aureum*). All identified polyphenols and other compounds, along with the molecular formulas and MS/MS data, are summarized in Appendix A and Appendix B. The polyphenols detected in our study were further categorized as flavones, flavonols, flavan-3-ols, anthocyanidins, phenolic acids, lignans, coumarins, stilbenes, etc. Overall, the metabolites detected in our study belonged to 54 compound classes. The highest number of polyphenols were flavonols (47), followed by anthocyanins (31), flavones (24), and flavan-3-ols (11) (Figure 1A,B). These numbers indicate that extracts of *R. pauciflorum*, *R. triste*, *R. dicuscha*, and *R. aureum* are rich in anthocyanins and flavonols. Among the other compound classes are pyranones, quinolines, amino acids, omega-3 fatty acids, terpenoids, and others (Appendix A and Appendix B). These results highlight that *Ribes* species’ berries are a rich source of a range of compounds. 

#### 2.1.1. Flavones

##### Hydroxy(iso)flavones

Two 7-hydroxyisoflavones, i.e., formononetin and luteolin-*O*-hexoside, were tentatively identified in the extracts from *R. aureum*, *R. triste*, and *R. pauciflorum*. The collision-induced (CID) spectrum in positive ion mode of formononetin from *R. aureum* is shown in Figure 2A. These compounds have been previously characterized as components of extracts from several plant species, including *Astragali Radix* [22], Huolisu oral liquid [23], *Dracocephalum jacutense* [24], *Medicago varia* [25], and *Maackia amurensis* [26]. The [M+H]^+^ ion produced two fragment ions with *m*/*z* 251.16 and *m*/*z* 137.27 (Figure 2A). The fragment ion with *m*/*z* 251.16 produced one characteristic daughter ion with *m*/*z* 233.30. The fragment ion with *m*/*z* 233.30 produced one characteristic daughter ion with *m*/*z* 150.11. 

##### Dihydroxyflavones

Our results also revealed the presence of the flavones acacetin, dihydroxy-methoxy(iso)flavone, cirsimaritin, dihydroxy-dimethoxy(iso)flavone, chrysoeriol 7-O-neohesperidoside, and chrysoeriol O-rhamnosyl glucoside in all four studied *Ribes* species. The CID spectrum in positive ion mode of acacetin from extracts of *R. triste* is shown in Figure 2B. These compounds have been previously characterized in extracts of *Rosmarinus officinalis* [27], propolis [28], and *M. varia* [25]. Acacetin has been previously reported in extracts from *Triticum aestivum* [29] and propolis [28]. The [M+H]^+^ ion produced three fragment ions with *m*/*z* 266.83, *m*/*z* 215.27, and *m*/*z* 133.08 (Figure 2B). The fragment ion with *m*/*z* 266.83 produced two characteristic daughter ions with *m*/*z* 241.05 and *m*/*z* 149.14. 

##### Trihydroxyflavones

The flavones apigenin, daidzin, vitexin, isovitexin, luteolin 7-*O*-(6-*O*-arabinosyl-glucoside), lonicerin, luteolin 7-*O*-(6-*O*-rhamnosyl-hexoside), kaempferol 3-*O*-(6-*O*-rhamnosyl-glucoside), and kaempferol 3-*O*-rutinoside have already been characterized as components of *Dryopteris ramosa* [30], *Aspalathus linearis* [31], *Artemisia annua* [32], lemon, passion fruit [33], and Phlomis (*Lamiaceae*) [34]. Trihydroxyflavones were tentatively identified in extracts from all four species of *Ribes.* The CID spectrum in negative ion mode of isovitexin from extracts of *R. aureum* is shown in Figure 2C. The [M-H]^−^ ion produced two fragment ions with *m*/*z* 283.28 and *m*/*z* 145.33 (Figure 2C). The fragment ion with *m*/*z* 145.33 produced one characteristic daughter ion with *m*/*z* 123.29. The mass spectrometry of isovitexin has been presented for extracts from *Rhus coriaria* [35], *Aspalathus linearis* [31], and Chilean currants [36].

##### Tetrahydroxyflavones

Among the tetrahydroxyflavones, kaempferol, isorhamnetin, rhamnetin II, quercetin 3-*O*-methyl ether, and quercitrin were tentatively identified in the extracts of all four *Ribes* species. The CID spectrum in positive ion mode of kaempferol from extracts of *R. pauciflorum* is shown in Figure 2D. The [M+H]^+^ ion produced one fragment ion with *m*/*z* 241.12 (Figure 2D). The fragment ion with *m*/*z* 145.33 produced one characteristic daughter ion with *m*/*z* 213.08. Of these compounds, kaempferol, an important compound in flavonoid biosynthesis and related pathways, has been reported in a diverse range of plant species, e.g., *Dryopteris ramosa* [30], *Inula gaveolens* [37], *Rhus coriaria* [29], *Juglans mandshurica* [38], *Lonicera japonica* [39], and *Ribes meyeri* [40]. Similarly, other tetrahydroxyflavones have been previously reported in *Inula gaveolens* [37], *Rhus coriaria L.* (*Sumac*) [29], *Spondias purpurea* [41], and *Polygala sibirica* [42]. 

##### Pentahydroxyflavones

Among other hydroxyflavones, pentahydroxyflavones such as quercetin, dihydroquercetin, and myricetin-3-*O*-galactoside were also present in the extracts of the *Ribes* species. These flavonols have been already characterized as components of *Juglans mandshurica* [38], *Vaccinium macrocarpon* [43], cranberry [44], and *Vaccinium myrtillus* [45]. The CID spectrum in positive ion mode of quercetin from extracts of *R. triste* is shown in Figure 3A. The [M+H]^+^ ion produced two fragment ions with *m*/*z* 257.12 and *m*/*z* 165.09 (Figure 3A). The fragment ion with *m*/*z* 257.12 produced two characteristic daughter ions with *m*/*z* 229.11 and *m*/*z* 201.13. The fragment ion with *m*/*z* 229.11 produced two characteristic daughter ions with *m*/*z* 201.08 and *m*/*z* 145.13.

#### 2.1.2. Flavan-3-ols

The catechins (epi)-catechin, afzelechin, gallocatechin, (epi)-gallocatechin, and (epi)-afzelechin-3-O-gallate have been already characterized as components of *Ribes meyeri* [42], *Ribes magellanicum* [36], *Vaccinium myrtillus* [45], *G. linguiforme* [46], and *Camellia kucha* [47]. The flavan-3-ols were tentatively identified in extracts from three species of *Ribes* (*R. dikuscha*, *R. pauciflorum*, *R. triste*). The CID spectrum in positive ion mode of gallocatechin from extracts of *R. triste* is shown in Figure 3B. The [M+H]^+^ ion produced two fragment ions with *m*/*z* 287.09 and *m*/*z* 153.11 (Figure 3B). The fragment ion with *m*/*z* 287.09 produced two characteristic daughter ions with *m*/*z* 259.10 and *m*/*z* 147.30. The fragment ion with *m*/*z* 259.10 produced one characteristic daughter ion with *m*/*z* 149.16. Gallocatechin is present in extracts of *G. linguiforme* [46], *Ribes meyeri* [42], *Vaccinium myrtillus* [48], and *Embelia* [49].

#### 2.1.3. Anthocyanins

The berries of all *Ribes* species showed unexpected enrichment in anthocyanins; 31 compounds from the anthocyanin group were identified. The berries of *Ribes dikuscha* were found to be the richest in the presence of anthocyanins. As an example, two mass spectra of the anthocyanin compounds identified in *R. dikuscha* extracts are presented below. The CID spectrum in negative ion mode of delphinidin 3,5-dihexoside from extracts of *R. dikuscha* is shown in Figure 3C. The [M–H]^−^ ion produced two fragment ions with *m*/*z* 299.78 and *m*/*z* 475.24 (Figure 3C). The fragment ion with *m*/*z* 299.78 produced one characteristic daughter ion with *m*/*z* 315.11. The anthocyanin delphinidin 3,5-dihexoside was tentatively identified in the literature in extracts from *F. herrerae* [46], *Berberis microphylla* [50], and Andean blueberry [51]. The CID spectrum in positive ion mode of petunidin-3-*O*-glucoside from extracts of *R. dikuscha* is shown in Figure 3D. The [M+H]^+^ ion produced one fragment ion with *m*/*z* 317.10 (Figure 3D). The fragment ion with *m*/*z* 317.10 produced one characteristic daughter ion with *m*/*z* 302.10. The fragment ion with *m*/*z* 302.10 produced one characteristic daughter ion with *m*/*z* 274.07. The mass spectrometry of petunidin-3-*O*-glucoside has been presented for extracts from black soybean [52], *Berberis ilicifolia* and *Berberis empetrifolia* [53], *Berberis microphylla* [50], grape [54], vines [55], and *Vigna angularis* [56].

### 2.2. Newly Detected Compounds in Genus Ribes

Of the detected metabolites in the four *Ribes* species, twenty-nine compounds from the polyphenol group and six compounds from other chemical groups were identified for the first time (Appendix A and Appendix B). The newly identified polyphenols include flavones, flavonols, flavan-3-ols, lignans, coumarin, stilbenes, etc. Moreover, some of the compound classes newly detected in *Ribes* species are the naphthoquinone group (1,8-dihydroxy-anthraquinone, 1,3,6,8-tetrahydroxy-9(10h)-anthracenone, 8,8′-dihydroxy-2,2′-binaphthalene-1,1′,4,4′-tetrone, etc.), polyhydroxycarboxylic acids, omega-3 fatty acids (stearidonic acid, linolenic acid), etc. (Figure 4A,B). 

The data obtained using the Venn diagram clearly show that seventeen polyphenolic compounds belonging to the groups of flavones, flavonols, anthocyanins, and phenolic acids are found in all four *Ribes* species. These include the polyphenols quercetin, 3,4-dihydroxyhydrocinnamic acid, apigenin, formononetin, kaempferol, ellagic acid, hydroxyferulic acid, lonicerin, syringaresinol, acacetin, bioquercetin, etc. (Appendix A; Figure 2B).

Furthermore, to identify the similarities and differences in the bioactive substances in different variations of *Ribes*, the Jaccard index was used (Table 1). Based on the polyphenolic compounds, the Jaccard index showed that *R. triste* and *R. aureum* are more similar, followed by *R. aureum* and *R. dikuscha* and by *R. triste* and *R. dikuscha.* The least similar species in terms of polyphenols were *R. aureum* and *R. pauciflorum*. 

Anthocyanins are an important class of flavonoids that give specific colors to berries, fruits, and plants. The four *Ribes* species also differed in their anthocyanin profiles. Most of these anthocyanins are being reported for the first time in *Ribes* species, indicating their predominant role in color formation in these species. The results showed that *R. triste* had the highest number of cyanidins and some delphinidins, corresponding to its bright red color. Meanwhile, the richest *Ribes* species in terms of anthocyanins was *R. dikuscha*; the highest number of anthocyanins were delphinidins, followed by petunidins, cyanidins, and malvidin, corresponding to their blackish blue berries. Similarly, *R. aureum* had delphinidins. However, this species had the lowest number of anthocyanins detected (Figure 4B). *R. triste* was the least similar to the other species in terms of anthocyanins (Table 2). Nevertheless, further dedicated research focusing on anthocyanins should reveal the detailed compositions of the studied *Ribes* berries. 

Finally, our data show that twelve polyphenolic compounds belonging to the groups flavones and flavonols were found in all four *Ribes* species. These are the flavonols quercetin, kaempferol, and bioquercetin and the flavones apigenin, formononetin, lonicerin, syringaresinol, acacetin, etc. (Figure 4C; Appendix A). The Jaccard index calculated for the sum of the flavone and flavonol compounds indicated that *R. aureum* and *R. triste* were relatively similar in terms of their flavone and flavonol compositions, whereas *R. aureum* and *R. pauciflorum* were the least similar (Table 3). 

## 3. Discussion

The genus *Ribes* (of the family *Grossulariaceae*) consists of more than 160 species. After strawberry, *Ribes* berries are preferred by consumers [57]. Among several *Ribes* species from Russia, four species, i.e., *R. dikuscha*, *R. pauciflorum*, *R. triste*, and *R. aureum*, are generally found in the Magadan region [16]. *Ribes* species are generally used for ethnomedical purposes in China and Russia. Several studies have highlighted their utility for the treatment of arthritis, joint pain, tuberculosis, hepatitis, gastrointestinal disorders, etc. This is because of their health-beneficial activity, such as anti-inflammatory, antioxidant, etc. [17]. Considering their increasing economic importance in the medicine, food, and dye industries, continued exploratory biochemical research is needed. To expand the scope of usage of *Ribes* species in the health and food industries, here, we explored the LC-MS profiles of four *Ribes* species. 

Metabolomic research on *Ribes* species is limited. Most work in this context has been dedicated to *Ribes nigrum* L. [58,59,60], whereas some articles also report *Ribes stenocarpum* Maxim. [61] and *Ribes fragrans* Pall. [62]. These and related research have highlighted that *Ribes* species’ berries are rich in polyphenols, which is consistent with our results (Figure 1). Work on *R. nigrum* has revealed the presence of the glucoside and rutinoside types of flavonols, such as myricetin, quercetin, kaempferol, and isorhamnetin [63]. In the case of the studied four *Ribes* species, *R. dikuscha*, followed by *R. triste* and *R. aureum*, had mostly rutinoside flavones, flavonols, and anthocyanins, whereas *R. pauciflorum* had no rutinosides. Similarly, the presence of glucosides (isoflavone, flavone, flavonol, and anthocyanins) indicates that *Ribes* berries are rich in glucoside and rutinoside polyphenols. Considering the broader role of polyphenols (e.g., quercetin, kaempferol, catechins, resveratrol, rutin), and their glycosides and rutinosides in particular, the use of the *Ribes* berries, therefore, should help in the prevention of various illnesses, e.g., cancer, cardiovascular diseases, diabetes, obesity, osteoporosis, liver-related diseases, neurodegenerative diseases, and a range of infections [64,65]. Of the four species, based on the number of detected polyphenols, *R. dikscha* offers a wider range of subclasses of compounds, followed by *R. triste*, *R. pauciflorum*, and *R. aureum*. However, a better conclusion can only be obtained based on further research on the content of each of these polyphenols. The differences observed in the metabolomic compositions of the berries of the four species indicate interspecific variation [66]. Interspecific metabolite composition differences can be due to a range of factors, such as the genetic background, variety [67,68], growing conditions [69], environmental conditions [70], agronomic practices, etc. Of the four *Ribes* species’ berries, *R. pauciflorum*’s berries were rich in flavan-3-ols (Appendix A). The rapidly growing body of literature and clinical data reflect their superior health benefits and lower risks [71]. Key benefits include improved blood pressure, sugar, and cholesterol levels [72]. In particular, the catechin, epi-catechin, and derivatives detected in *R. pauciflorum* berries are useful as these compounds offer therapeutic benefits in inflammatory bowel disease [73], UV protection, and inflammation inhibition, as well as in acne, neurodegeneration, and several other diseases [74]. Based on these results, future research should be conducted on their quantitative determination and their health benefits. Taken together, our results present the compositions of four *Ribes* species and highlight key similarities and differences. 

One of the most bioactive classes of compounds in the *Ribes* species studied is anthocyanins [60,75]. This class of polyphenols contributes to the color, aroma, taste, and astringency of the fruits and berries [75,76]. Work on *R. nigrum* has indicated the presence of cyanidins, delphinidins, pelargonidins, peonidins, and cyanidins [75]. Consistent with the fact that it had the highest number of detected polyphenols, the highest number and range of anthocyanins found in *R. dikuscha* indicate that the berries of this species would be of better use in the food and pharmaceutical industries. Moreover, the fact that glucosides, beta-galactosides, dihexosides, hexosides, hexuronides, rutinosides, etc., were detected is indicative of the presence of diverse anthocyanins (Appendix A). These observations offer opportunities for interspecific hybridization to improve the anthocyanin compositions of other *Ribes* species. 

*Ribes* species also contain other classes of compounds, including organic acids, flavoring components, essential oils, polysaccharides, and others, e.g., biphenyls, nitrile-containing compounds, lignans, terpenoids, etc. [17]. In this regard, the detection of a range of compounds belonging to classes such as amino acids, quinones, carboxylic acids, omega-3 fatty acids, terpenoids, fatty acids, etc., is an important finding. Notably, the detection of L-theanine in *R. aureum* suggests that the consumption of its berries might offer anti-anxiety, stress-relieving, and insomnia-reducing effects [77]. Meanwhile, the presence of terpenoids such as cryptotanshinone, pregnane-3,11,17,20-tetrol, lup-2,20(29)-dien-28-ol, and sespendole is consistent with the results reported for *R. nigrum* [78]. However, their detection in the studied species highlights that terpenoids are prevalent in *Ribes* berries. These results also imply that continued research on other *Ribes* species is required to broaden the utility of these species. The number of terpenoids detected in this work is small compared to those reported in *R. nigrum* because of the different technologies used [18,19,79]. Other techniques, e.g., those based on MS, should be employed, and a complete understanding of the composition of *Ribes* species’ volatilome should be obtained. Taken together, our results indicate that the studied *Ribes* species’ metabolome is diverse and compounds other than polyphenols are prevalent in their berries. 

## 4. Materials and Methods

### 4.1. Materials

The objects of this study were the berries of *Ribes* species (*Ribes pauciflorum* Turcz., *Ribes triste* Pall., *Ribes dicuscha* Fisch.). The plant material was collected in two expeditions during July 2023 and June 2024 to the Magadan region (Russian Federation). Samples of the collected expedition material and a plant collection map are presented in Figure 5. The species *Ribes aureum* Purch. was obtained from a plantation at the Far-Eastern Branch of the N.I. Vavilov All-Russian Institute of Plant Genetic Resources.

Triplicate samples were collected for each accession/variety. Care was taken to collect healthy, disease- and insect-free berries. Samples were washed with distilled water and stored at −80 °C until processing. All samples morphologically corresponded to the pharmacopeial standards of the State Pharmacopoeia of the Russian Federation [80].

### 4.2. Chemicals and Reagents

HPLC-grade acetonitrile was purchased from Fisher Scientific (Kent, UK), and MS-grade formic acid was purchased from Sigma-Aldrich (Steinheim, Germany). Ultrapure water was prepared from a SIEMENS Ultra-Clear system (SIEMENS Water Technologies, Munich, Germany), and all other chemicals were of analytical grade.

### 4.3. Extraction

The fractional maceration technique was used to obtain highly concentrated extracts. Aqueous ethanol was used for extraction. Here, 50 g of berries of each species was randomly selected for maceration. The total amount of the extractant (aqueous ethanol 95%) was divided into three parts, and the parts of the plant were consistently infused with the first, second, and third parts. The infusion of each part of the extractant lasted seven days at room temperature. Three replicates of the extraction process were carried out on each plant sample. The extract was filtered through Whatman filter paper. The filtrates were diluted with acetonitrile to the final working concentration for analysis.

### 4.4. Liquid Chromatography

HPLC was performed using a Shimadzu LC-20 Prominence HPLC device (Shimadzu, Kyoto, Japan) equipped with a UV sensor and a C18 silica reverse-phase column (4.6 × 150 mm, particle size: 2.7 μm) to perform the separation of the multicomponent mixtures. The gradient elution program, with two mobile phases (A, deionized water; B, acetonitrile with formic acid 0.1% *v*/*v*), was as follows: 0–2 min, 0% B; 2–50 min, 0–100% B; control washing 50–60 min 100% B. The entire HPLC analysis was performed with an ESI detector at a wavelength of 230 nm for the identification of compounds; the temperature was 50 °C, and the total flow rate was 0.25 mL min^−1^. The injection volume was 10 μL. Additionally, liquid chromatography was combined with a mass-spectrometric ion trap to identify compounds.

### 4.5. Mass Spectrometry

MS analysis was performed on an ion trap amaZon SL (BRUKER DALTONIKS, Bremen, Germany) equipped with an ESI source in negative ion mode. The optimized parameters were obtained as follows: ionization source temperature, 70 °C; gas flow, 4 L/min; nebulizer gas (atomizer), 7.3 psi; capillary voltage, 4500 V; end plate bend voltage, 1500 V; fragmentary, 280 V; collision energy, 60 eV. An ion trap was used in the scan range *m*/*z* 100–1700 for MS and MS/MS. The chemical constituents were identified by comparing their retention indices, mass spectra, and MS fragmentation with an in-house self-built database (Biotechnology, Bioengineering and Food Systems Laboratory, Far-Eastern Federal University, Russia). The in-house self-built database was based on data from other spectroscopic techniques, such as nuclear magnetic resonance, ultraviolet spectroscopy, and MS, as well as data from the literature, and it is continuously updated and revised. The capture rate was one spectrum for MS and two spectra for MS/MS. Data acquisition was controlled by the Metaboscape for BRUKER DALTONIKS. All experiments were repeated three times. A four-stage ion separation mode (MS/MS mode) was implemented.

### 4.6. Data Analysis and Visualization

Venn diagrams were prepared using an online tool, InteractiVenn [81]. Bar plots for the compound classes and number of compounds in each class were prepared in Microsoft Excel 2021 Professional^®^ (www.microsoft.com). The scatter plot of the detected compounds was prepared in TBtools [21].

To present the similarities and differences in the bioactive substances in different variations of *Ribes*, the Jaccard index (Jaccard similarity coefficient) [82] was used, which evaluates the similarity and diversity of sets of samples. The Jaccard index measures the similarity between finite sample sets and is defined as the size of the intersection divided by the size of the union of the sample sets:$$\left(A,B\right)=\frac{\left|A\cap B\right|}{\left|A\right|+\left|B\right|-|A\cap B|}$$

Note that, by design, $0\leq J\left(A,B\right)\leq 1.$

## 5. Conclusions

Our results reveal that the berries of the *Ribes* species (*R. dikuscha*, *R. pauciflorum*, *R. triste*, *R. aureum*) contain a range of polyphenolic and non-polyphenolic constituents; two hundred and five different bioactive components have been identified in *Ribes* extracts. Several polyphenols were putatively detected for the first time in the studied *Ribes* species. These findings indicate that *R. dikuscha* berries are richer in polyphenols and anthocyanins compared to those of the other three species. *R. pauciflorum* is rich in flavan-3-ols. The polyphenols of the *Ribes* species also included glucosides and rutinosides. The *Ribes* berries’ metabolomes also consist of compounds belonging to classes such as amino acids, quinones, carboxylic acids, omega-3 fatty acids, terpenoids, fatty acids, etc. The comparative analysis highlights the presence of interspecific polyphenolic and anthocyanin differences in *Ribes* species. These results are highly important in expanding the utility of these species in the food and pharmaceutical industries. 

## Figures and Tables

**Figure 1 ijms-25-10085-f001:**
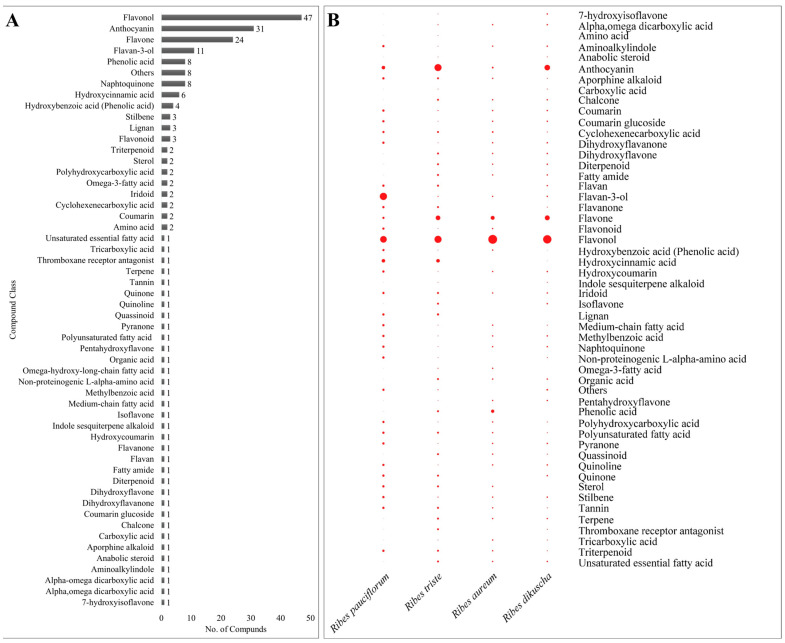
Global metabolome profile of *Ribes* species. (**A**) No. of compounds detected in each compound class in all species. (**B**) No. of compounds detected in each *Ribes* species. The circle size indicates the number of compounds. (**B**) was prepared in TBtools [21].

**Figure 2 ijms-25-10085-f002:**
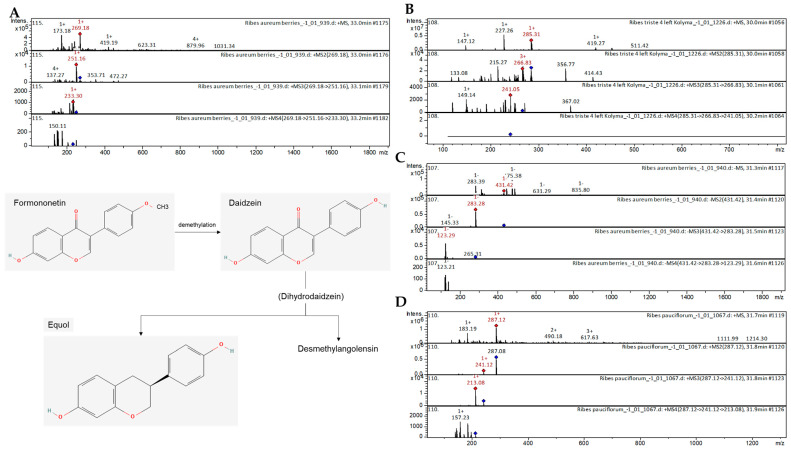
(**A**) CID spectrum of formononetin from *R. aureum*, *m*/*z* 269.18. The chemical structures correspond to panel A of the figure. The formulas were obtained from the PubChem database of the National Library of Medicine, National Center for Biotechnology Information. (**B**) CID spectrum of acacetin from *R. triste*, *m*/*z* 285.31. (**C**) CID spectrum of isovitexin from extract of *R. aureum*, *m*/*z* 431.42. (**D**) CID spectrum of kaempferol from extract of *R. pauciflorum*, *m*/*z* 287.12.

**Figure 3 ijms-25-10085-f003:**
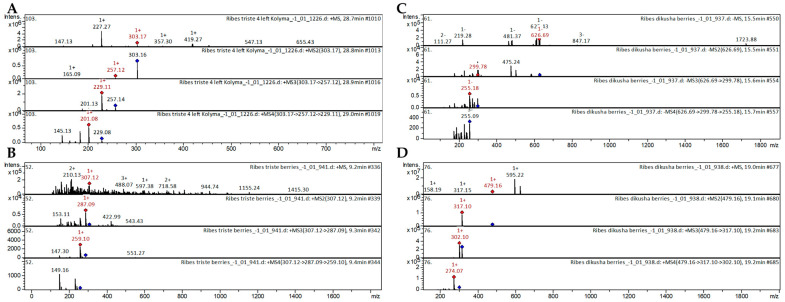
(**A**) CID spectrum of quercetin from extract of *R. triste*, *m*/*z* 303.17. (**B**) CID spectrum of gallocatechin from extracts of *R. triste*, *m*/*z* 307.12. (**C**) CID spectrum of anthocyanin delphinidin 3,5-dihexoside from berries of *R. dikuscha*, *m*/*z* 626.69. (**D**) CID spectrum of anthocyanin petunidin-3-*O*-glucoside from berries of *R. dikuscha*, *m*/*z* 479.16.

**Figure 4 ijms-25-10085-f004:**
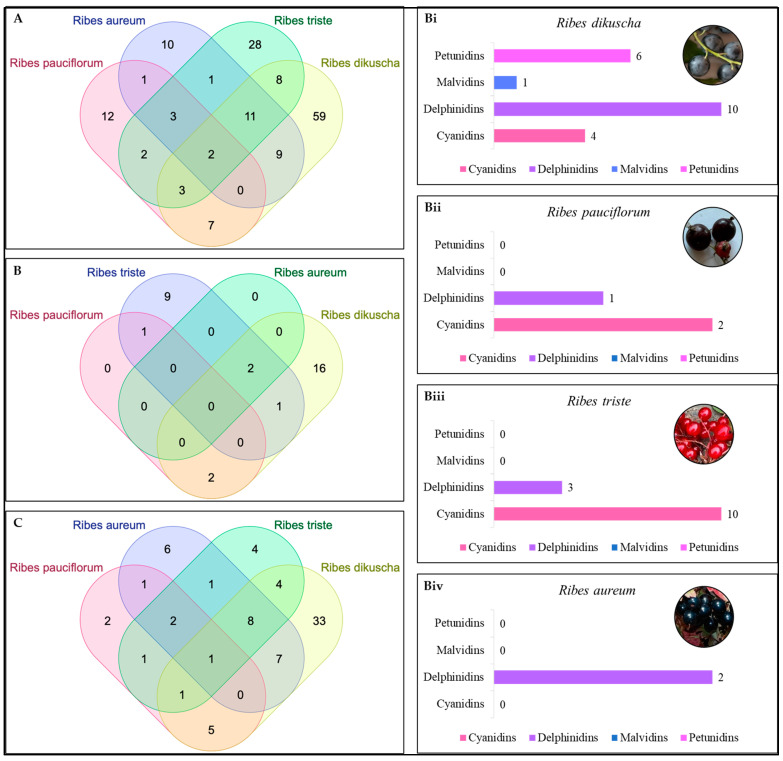
(**A**) Venn diagram showing similarities and differences in the presence of the polyphenol group, (**B**) anthocyanins, and (**C**) flavones and flavonols in *Ribes* species. Bi-iv panels (x-axis) indicate the type of anthocyanins detected in each *Ribes* species.

**Figure 5 ijms-25-10085-f005:**
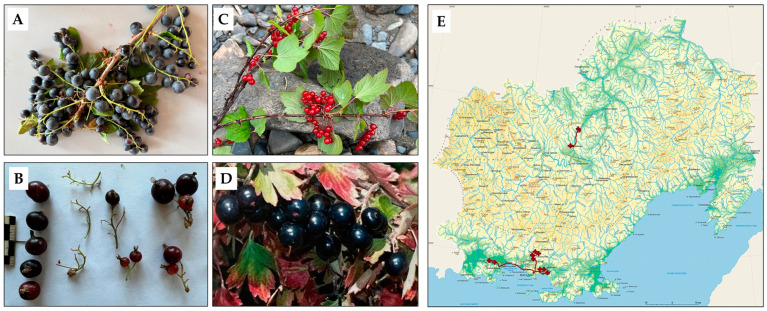
Plant samples used in this study. (**A**) *Ribes dicuscha* (the vicinity of the Kolyma River, N = 59°4141′960 E = 151°16′17.620). (**B**) *Ribes pauciflorum*, (**C**) *Ribes triste*, (**D**) *Ribes aureum* (Seymchansky District, the vicinity of the Kolyma River, N = 62°55′51.017 E = 151°16′17.620. (**E**) Map of the route and collection of plant material in the Magadan region, Russian Federation (N 58–61°, E 150–153°).

**Table 1 ijms-25-10085-t001:** Jaccard index for four *Ribes* species and the polyphenol group (*R. pauciflorum*, *R. aureum*, *R. triste*, *R. dikuscha*).

	*R. pauciflorum* −30	*R. aureum* −37	*R. triste* −58	*R. dikuscha* −99
*R. pauciflorum* −30		6 0.0984	10 0.1282	12 0.1026
*R. aureum* −37	6 0.0984		17 0.2179	22 0.193
*R. triste* −58	10 0.1282	17 0.2179		24 0.1805
*R. dikuscha* −99	12 0.1026	22 0.193	24 0.1805	

**Table 2 ijms-25-10085-t002:** Jaccard index for four species of *Ribes* (anthocyanin group).

	*R. pauciflorum* −3	*R. aureum* −13	*R. triste* −2	*R. dikuscha* −21
*R. pauciflorum* −3		1 0.0667	0 0	2 0.0909
*R. aureum* −13	1 0.0667		2 0.1538	3 0.0968
*R. triste* −2	0 0	2 0.1538		2 0.0952
*R. dikuscha* −21	2 0.0909	3 0.0968	2 0.0952	

**Table 3 ijms-25-10085-t003:** Jaccard index for four species of *Ribes* (flavones and flavonols).

	*R. pauciflorum* −13	*R. aureum* −26	*R. triste* −22	*R. dikuscha* −59
*R. pauciflorum* −13		4 0.1143	5 0.1667	7 0.1077
*R. aureum* −26	4 0.1143		12 0.3333	16 0.2319
*R. triste* −22	5 0.1667	12 0.3333		14 0.209
*R. dikuscha* −59	7 0.1077	16 0.2319	14 0.209	

## Data Availability

The original contributions presented in the study are included in the article; further inquiries can be directed to the corresponding author/s.

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
