# Peer review of "Genus Ribes: Ribes aureum, Ribes pauciflorum, Ribes triste, and Ribes dikuscha—Comparative Mass Spectrometric Study of Polyphenolic Composition and Other Bioactive Constituents"

_ijms, 2024, doi:10.3390/ijms251810085_

Round 1

Reviewer 1 Report

Comments and Suggestions for Authors

In this work tandem mass-spectrometry was applied to tentatively identified 205 bioactive compounds (155 19 compound of polyphenol group and 50 compounds from other chemical groups) in extracts from 20 berries of four species of Ribes.

The text needs careful revision in terms of writing (repetitions, incomplete lists, poor description of results, and poor table comments) and text format (including tables and figures).

Section 3 has a list of compounds, some of which are accompanied by spectra. All compounds and fragments are given in Appendix 1, Table 1. By what criteria were the compounds listed in Section 3 selected? This part also gives repetitions in terms of bibliographical references.

Section 4 is full of tables and graphs but lacks adequate and thorough discussion of these.

For all these reasons I suggest the publication of the article after a major revision.

Minor comments:

·        The abstract is just a list of compounds and should be modified including some more relevant information.

·        Line 80: check the verb “were was”

·        Figure 2 caption: missing closing parenthesis

·        Line 104: the extraction solvent in not clear, 80% or 95% (line 106) of aqueous ethanol?

·        Line 118: UV-Vis was use “at a wavelength of 230 nm for identification compounds”? The identification was not performed with MS?

·        It is not clear from tables 1 and 2 (Appendix 1) whether these compounds were found for each Ribes species or whether they are common to all species.

·        Section 3.2.1.1: what about luteolin-O-hexoside?

·        Lines 167-168: this is a repetition of lines 157-158.

·        Section 3.3 is missing?

·        Lines 275-258: an exhaustive list is needed without etc. Perhaps in the form of a table

·        Figure 12 and Table 1 captions: Vienna or Venn diagram? Be consistent.

·        Lines 288-290: an incomplete list is not useful if it is then listed exhaustively in a table. You could change the sentence to “the list of polyphenols is reported in table1”.

·        Lines 299-300: to aid understanding of the table, a numerical explanation of the Jaccard index could be added: its value is between 0 and 1, as correctly explained in the text, but what does it mean if the value approaches 0 or 1? Also missing is a commentary regarding the indices obtained and shown in Table 2. The same applies to the following tables.

·        The conclusions need to be reworded, removing the incomplete list of compounds, but perhaps focusing on which compounds are most present, which are common to all species examined, and which are exclusive to a single species. Is the same for the Abstract: less list of compounds and more relevant results.

Author Response

Authors response to Reviewer 1 comments

Dear reviewer 1, thank you for your valuable time spend on revision of our submission. We have now revised the manuscript in light of your comments and significantly improved it. We used track-changes function in Microsoft word. A point-by-point response is appended below.

Reviewer 1: The text needs careful revision in terms of writing (repetitions, incomplete lists, poor description of results, and poor table comments) and text format (including tables and figures).

Authors response: Dear reviewer 1, we have now revised the text from all perspectives.

The language has been improved in terms of grammar and presentation.

Formatting of tables and lists has been carefully checked and updated accordingly.

Writing mistakes and repetitions have been corrected and improved.

Reviewer 1: Section 3 has a list of compounds, some of which are accompanied by spectra. All compounds and fragments are given in Appendix 1, Table 1. By what criteria were the compounds listed in Section 3 selected? This part also gives repetitions in terms of bibliographical references.

Authors response: Dear reviewer 1, we have now updated the section 3 (now section 2). The section 2 presents all the results on the detection of compounds in the four species. We have presented them according to classes. However, large scale changes in the way, the compounds are presented as results have been done. For example, figure 1 has been added highlighting the major compound classes detected and number of compounds in each class. Figure 1B has been added for individual species. Similarly, figure 2 and figure 3 have been compiled. Figure 4 has been added on the anthocyanins. Moreover, a dedicated Discussion section has been added. Regarding bibliography, care has been taken. However, we tended to use bibliography when the compound is presented in results and if it has been previously detected in other crop plants.

Reviewer 1: Section 4 is full of tables and graphs but lacks adequate and thorough discussion of these.

For all these reasons I suggest the publication of the article after a major revision.

Reviewer 1: Minor comments:

  • The abstract is just a list of compounds and should be modified including some more relevant information.

Abstract has been modified. The list of compounds is removed instead a more concise but comprehensive abstract is presented

  • Line 80: check the verb “were was”

Manuscript has been checked for English language.

  • Figure 2 caption: missing closing parenthesis

Figure captions have been updated.

  • Line 104: the extraction solvent in not clear, 80% or 95% (line 106) of aqueous ethanol?

Text has been updated. It is aqueous ethanol (95%).

  • Line 118: UV-Vis was use “at a wavelength of 230 nm for identification compounds”? The identification was not performed with MS?

Dear Editors! An unfortunate technical error occurred here. The text has been corrected.

The entire HPLC analysis was carried out with an ESI detector at wavelengths of 230 and 330 nm; the temperature was set to 17 °C, and the injection volume was 0.25 mL min-1.

  • It is not clear from tables 1 and 2 (Appendix 1) whether these compounds were found for each Ribes species or whether they are common to all species.

Dear Editor! The team of authors has inserted additional table into the Appendix A and B which indicates the compounds presences in Ribes specie(s).

  • Section 3.2.1.1: what about luteolin-O-hexoside?

Dear reviewer, what we understand from the comment is that you asked about the CID-spectrum of the luteolin-o-hexoside. For brevity, we have opted to present CID-spectrum for only 8 compounds in the MS.

  • Lines 167-168: this is a repetition of lines 157-158.

Dear reviewer, the text has been improved and corrected for repetitions.

  • Section 3.3 is missing?

Dear reviewer, the section numbering has been checked and corrected. Moreover, the whole MS has been rearranged as advised by the editor i.e., Intro, Results, Discussion, Conclusion, and Methods. The sub-section numbering has also been updated.

  • Lines 275-258: an exhaustive list is needed without etc. Perhaps in the form of a table

Dear reviewer, the list of compounds is now provided only in Appendix A and B and major classification and compound numbers are given in figures. By doing so, we tried to present key information and not the redundant information.

  • Figure 12 and Table 1 captions: Vienna or Venn diagram? Be consistent.

Dear reviewer, it is Venn diagram. We have now checked and corrected.

  • Lines 288-290: an incomplete list is not useful if it is then listed exhaustively in a table. You could change the sentence to “the list of polyphenols is reported in table1”.

Dear reviewer, the list of compounds is now provided only in Appendix A and B and major classification and compound numbers are given in figures. By doing so, we tried to present key information and not the redundant information.

  • Lines 299-300: to aid understanding of the table, a numerical explanation of the Jaccard index could be added: its value is between 0 and 1, as correctly explained in the text, but what does it mean if the value approaches 0 or 1? Also missing is a commentary regarding the indices obtained and shown in Table 2. The same applies to the following tables.

Dear reviewer, we have explained the results of Jaccard index for each category. Thank you for the comment.

  • The conclusions need to be reworded, removing the incomplete list of compounds, but perhaps focusing on which compounds are most present, which are common to all species examined, and which are exclusive to a single species. Is the same for the Abstract: less list of compounds and more relevant results.

Dear reviewer, Thank you for the comment. Conclusion has been updated. In general, we have significantly improved the whole manuscript in terms of presentation, style, figures, tables, and text.

Reviewer 2 Report

Comments and Suggestions for Authors

This study investigates the metabolomic composition of the polyphenolic compounds in four blackcurrant species from the genus Ribes using mass spectrometry. A total of 205 bioactive compounds were identified in the berry extracts of these species, including 25 polyphenolic constituents identified for the first time. This research provides valuable insights into the composition of these four Ribes species. However, there are several areas that need improvement:

(1) In the Introduction section, the significance of the study is unclear. It is not evident why only mass spectrometry was used to explore the composition of these Ribes species. The introduction should clarify the progress in related research and specify the types of polyphenolic compounds identified in the berry extracts.

(2) Continued to the above comment, the last paragraph of the Introduction should clearly outline what has been investigated in this study.

(3) The manuscript contains too many Figures and Tables. Please retain only the most crucial figures and tables in the main text and move the rest to the Supporting Information.

(4) Appendix 1, Table 1 shows that only low-resolution mass spectrometry and tandem mass spectra were used. For more accurate structure confirmation, high-resolution mass spectrometry should be employed.

(5) When identifying related compounds, include the fragment composition. For instance, in the discussion of the CID spectrum of Formononetin from R. aureum, the fragment ions m/z 269.18, 251.16, and 233.30 should be detailed with their corresponding structures.

(6) The significance of the Jaccard Index for the four species needs further explanation. Tables 2-5 list the Jaccard Index, but their implications are not discussed.

(7) The Conclusions should highlight the significance of the findings of this study.

Comments on the Quality of English Language

n/a

Author Response

Authors response to Reviewer 2 comments

Reviewer 2: This study investigates the metabolomic composition of the polyphenolic compounds in four blackcurrant species from the genus Ribes using mass spectrometry. A total of 205 bioactive compounds were identified in the berry extracts of these species, including 25 polyphenolic constituents identified for the first time. This research provides valuable insights into the composition of these four Ribes species. However, there are several areas that need improvement:

Authors response: Dear reviewer 2, thank you for your valuable comments and time spent on evaluation of our submission. We have now significantly improved the manuscript in terms of quality of presentation, figures, tables, text. Moreover, an English language check has been done. All the changes made to the text are given using Track-changes option of the Microsoft Word. A point-by-point response to your comments is given below.

Reviewer 2: (1) In the Introduction section, the significance of the study is unclear. It is not evident why only mass spectrometry was used to explore the composition of these Ribes species. The introduction should clarify the progress in related research and specify the types of polyphenolic compounds identified in the berry extracts.

Authors response: Dear reviewer 2, we have now improved the introduction section and added the significance of the species and why it is important to study the material. Please see the revised version of the manuscript section 1. Introduction.

Reviewer 2: (2) Continued to the above comment, the last paragraph of the Introduction should clearly outline what has been investigated in this study.

Authors response: Dear reviewer 2, yes, you are exactly right. We have now updated the introduction section. Especially added a paragraph at the end of the section.

Reviewer 2: (3) The manuscript contains too many Figures and Tables. Please retain only the most crucial figures and tables in the main text and move the rest to the Supporting Information.

Authors response: Dear reviewer 2, yes, we are aware of this mis-representation of the figures. We have compiled the important information to be provided as figures. Now the manuscript contains only five figures.

Reviewer 2: (4) Appendix 1, Table 1 shows that only low-resolution mass spectrometry and tandem mass spectra were used. For more accurate structure confirmation, high-resolution mass spectrometry should be employed.

Authors response: Dear Reviewer. In our studies, we used High Performance Liquid Chromatography + ion trap. This method gives an tentative determination of chemical compounds, which we always write about. This method is used to determine chemical compounds in many scientific articles. The most accurate method of determination is, of course, nuclear magnetic resonance. But it is not used in this article.

Reviewer 2: (5) When identifying related compounds, include the fragment composition. For instance, in the discussion of the CID spectrum of Formononetin from R. aureum, the fragment ions m/z 269.18, 251.16, and 233.30 should be detailed with their corresponding structures.

Authors response: Dear reviewer we have provided the corresponding structures as an example for the first compound we presented. Please see revised manuscript Figure 2A.

Reviewer 2: (6) The significance of the Jaccard Index for the four species needs further explanation. Tables 2-5 list the Jaccard Index, but their implications are not discussed.

Our response: Dear reviewer, we have explained the results of Jaccard index for each category. Thank you for the comment.

Reviewer 2: (7) The Conclusions should highlight the significance of the findings of this study.

Our response: Dear reviewer, Thank you for the comment. Conclusion has been updated. In general, we have significantly improved the whole manuscript in terms of presentation, style, figures, tables, and text.

Round 2

Reviewer 1 Report

Comments and Suggestions for Authors

The personal form is discouraged in the writing of scientific articles so it is suggested to replace the personal form “ we report” with the impersonal “the study or work reports...”, throughout the text, including abstracts.

In the previous review it was pointed out that there was a list of substances in the abstract. In spite of the changes made by the authors, I believe that “etc.” is not acceptable: it would be better to have a complete list or a list that covers only certain compounds of interest.

Careful re-reading of the text after accepting all revisions is recommended: by way of illustration I point out that there is still a repetition left at line 490 “in extracts in the extracts”, capital letter for "Formononetin" in Fig 2 caption, and other.

Line 228: “The structural identification of each compound was performed on the basis of their accurate mass and MS/MS fragmentation by HPLC-ESI-ion trap-MS/MS.” Were standards or literature data used? In the former case, the std solutions used should also be added in materials and methods.

Author Response

Authors responses to Reviewer 1

Reviewer 1. The personal form is discouraged in the writing of scientific articles so it is suggested to replace the personal form “ we report” with the impersonal “the study or work reports...”, throughout the text, including abstracts.

Our response. Dear reviewer 1, we have now corrected the manuscript regarding the use of personal form. Please refer to the updated manuscript for the said changes.

Reviewer 1. In the previous review it was pointed out that there was a list of substances in the abstract. In spite of the changes made by the authors, I believe that “etc.” is not acceptable: it would be better to have a complete list or a list that covers only certain compounds of interest.

Our response. Dear reviewer 1, we have now corrected the abstract regarding the use of etc. Please refer to the updated manuscript for the said changes.

Reviewer 1. Careful re-reading of the text after accepting all revisions is recommended: by way of illustration, I point out that there is still a repetition left at line 490 “in extracts in the extracts”, capital letter for "Formononetin" in Fig 2 caption, and other.

Our response. Dear reviewer 1, we have updated the manuscript regarding the repetition you highlighted, use of capital first letter for compounds in figure 2 caption, abstract, and others.

Reviewer 1. Line 228: “The structural identification of each compound was performed on the basis of their accurate mass and MS/MS fragmentation by HPLC-ESI-ion trap-MS/MS.” Were standards or literature data used? In the former case, the std solutions used should also be added in materials and methods.

Our response. Dear reviewer 1, the chemical constituents were identified by comparing their retention index, mass spectra, and MS fragmentation with an in-house self-built database (Biotechnology, Bioengineering and Food Systems Laboratory, Far-Eastern Federal University, Russia).

Reviewer 2 Report

Comments and Suggestions for Authors

All of my concerns have been addressed. Congratulations!

Comments on the Quality of English Language

n/a

Author Response

Dear editor,

International Journal of Molecular Sciences,

Submitted for your consideration is the revised version of our manuscript entitled “Genus Ribes: Ribes aureum, Ribes pauciflorum, Ribes triste, and Ribes dikuscha, comparative mass spectrometric study of poly-phenolic composition and other bioactive constituents”.

We have now further improved the manuscript in light of the comments from the reviewer 1. The changes are given using track-changes function in the Microsoft world. A point-by-point response is appended below.

We are hopeful that the revised version is a much-improved manuscript and will become a useful reference point for the community working on Ribes species.

Best regards